# Inhibition of Transglutaminase 2 as a Therapeutic Strategy in Celiac Disease—In Vitro Studies in Intestinal Cells and Duodenal Biopsies

**DOI:** 10.3390/ijms24054795

**Published:** 2023-03-01

**Authors:** Sebastian Stricker, Jan de Laffolie, Klaus-Peter Zimmer, Silvia Rudloff

**Affiliations:** 1Department of Pediatrics, Justus-Liebig-University Giessen, 35392 Giessen, Germany; 2Institute of Nutritional Science, Justus-Liebig-University Giessen, 35392 Giessen, Germany

**Keywords:** celiac disease, transglutaminase 2, TG2 inhibitor, PX-12, gliadin transport

## Abstract

Enzymatic modification of gliadin peptides by human transglutaminase 2 (TG2) is a key mechanism in the pathogenesis of celiac disease (CD) and represents a potential therapeutic target. Recently, we have identified the small oxidative molecule PX-12 as an effective inhibitor of TG2 in vitro. In this study, we further investigated the effect of PX-12 and the established active-site directed inhibitor ERW1041 on TG2 activity and epithelial transport of gliadin peptides. We analyzed TG2 activity using immobilized TG2, Caco-2 cell lysates, confluent Caco-2 cell monolayers and duodenal biopsies from CD patients. TG2-mediated cross-linking of pepsin-/trypsin-digested gliadin (PTG) and 5BP (5-biotinamidopentylamine) was quantified by colorimetry, fluorometry and confocal microscopy. Cell viability was tested with a resazurin-based fluorometric assay. Epithelial transport of promofluor-conjugated gliadin peptides P31-43 and P56-88 was analyzed by fluorometry and confocal microscopy. PX-12 reduced TG2-mediated cross-linking of PTG and was significantly more effective than ERW1041 (10 µM, 15 ± 3 vs. 48 ± 8%, *p* < 0.001). In addition, PX-12 inhibited TG2 in cell lysates obtained from Caco-2 cells more than ERW1041 (10 µM; 12 ± 7% vs. 45 ± 19%, *p* < 0.05). Both substances inhibited TG2 comparably in the intestinal lamina propria of duodenal biopsies (100 µM, 25 ± 13% vs. 22 ± 11%). However, PX-12 did not inhibit TG2 in confluent Caco-2 cells, whereas ERW1041 showed a dose-dependent effect. Similarly, epithelial transport of P56-88 was inhibited by ERW1041, but not by PX-12. Cell viability was not negatively affected by either substance at concentrations up to 100 µM. PX-12 did not reduce TG2 activity or gliadin peptide transport in confluent Caco-2 cells. This could be caused by rapid inactivation or degradation of the substance in the Caco-2 cell culture. Still, our in vitro data underline the potential of the oxidative inhibition of TG2. The fact that the TG2-specific inhibitor ERW1041 reduced the epithelial uptake of P56-88 in Caco-2 cells further strengthens the therapeutic potential of TG2 inhibitors in CD.

## 1. Introduction

Celiac disease (CD) is one of the most common autoimmune disorders, affecting about 1% of the Western population [1]. The disease is initiated by inadequately digested prolamine peptides that trigger an immune response in individuals carrying the predisposing HLA DQ2/8 genotype [2]. Clinically, CD can present with multifaceted symptoms, including those in the gastrointestinal tract (e.g., bloating and diarrhea), but also at extraintestinal sites (e.g., failure to thrive and anemia) [3]. Despite extensive research efforts, a strict and life-long gluten-free diet is the only available therapy to date. It represents an effective and safe treatment when conducted properly (i.e., well-balanced and covering all nutritional requirements). Still, the diet is a major burden for patients and their families, as it requires complete abstinence from all common bakery products, pasta and convenience products containing gluten or contaminations. Eating out further represents an organizational challenge, and a high number of patients, especially adolescents and adults, do not adhere properly to the gluten-free diet [4,5]. CD is associated with moderately increased morbidity and mortality due to disease complications, such as malabsorption, osteoporosis and the enteropathy-associated T-cell lymphoma [6,7]. Strict adherence to a gluten-free diet can mitigate those complications [8]. However, a gluten-free diet can also be associated with a lower intake of essential nutrients (magnesium, iron, zinc, manganese and folate) and a higher intake of dietary fat, leading to negative health consequences [9,10,11]. These facts indicate the high need for non-dietary treatment options in CD.

One potential therapeutic target is transglutaminase 2 (EC 2.3.2.13, TG2), the central autoantigen of the disease [12]. TG2 is ubiquitously expressed and belongs to a family of eight calcium-dependent enzymes (TG1-7 and blood coagulation factor XIII). TG2 expression is induced by multiple events, including apoptosis, endoplasmic reticulum stress, tissue remodeling and inflammation (IFN-γ and NF-κB). The enzyme can be activated by high calcium levels (cell damage and endoplasmic reticulum stress) and plays a role in multiple biological processes, including cell growth, differentiation and tissue repair [13,14]. TG2 deamidates specific epitopes of α-gliadin (PQLP→PELP), which significantly increases their immunogenicity [15,16,17]. This enzymatic modification is a precondition for the subsequent antigen presentation that initiates the Th1-mediated immune response. This leads to the production of autoantibodies (anti-TG2-IgA-Ab) and inflammatory cytokines (IFN-γ), finally resulting in the destruction of the intestinal mucosa [18]. In addition to its well-known role in the intestinal lamina propria, TG2 also takes part in endocytotic recycling and transcytosis [19,20,21]. 

Inhibition of TG2 was implicated as a therapeutic target in CD, and the potential of competitive TG2 inhibitors has been shown by in vitro studies and a single phase-II clinical trial [22,23]. In addition, the oxidative regulation of TG2 by endogenous proteins has been shown, as well [24,25]. Recently, we have demonstrated on an in vitro level that the oxidative inhibition of TG2 is much more effective than competitive inhibition, and we identified the small oxidative molecule PX-12 (2-[(1-methylpropyl)dithio]-1H-imidazole) as a potential drug candidate in CD [26]. In this study, we further investigated the effect of PX-12 and the established, active-site directed inhibitor ERW1041 on TG2 activity and epithelial transport of gliadin peptides. 

## 2. Results

### 2.1. PX-12 Inhibits Transamidation of Gliadin More Effectively Than ERW1041

First, we wanted to address the question whether PX-12 and ERW1041 are effective in reducing the TG2-mediated transamidation of gliadin. Therefore, we investigated the cross-linking of commercially available digested gliadin (PTG, T004, Zedira, Darmstadt, Germany) by TG2. The coating of the cavities on a 96-well plate with 10 nM TG2 resulted in a prominent increase in detected PTG compared to the direct adherence of PTG (100 ± 9 vs. 7 ± 7%, respectively; *p* < 0.0001, Figure 1A). Both inhibitors, PX-12 and ERW1041, reduced TG2-mediated cross-linking of PTG in a dose-dependent manner. At tested concentrations of both 10 µM (15 ± 3 vs. 48 ± 8%, *p* < 0.001) and 100 µM (5 ± 1 vs. 11 ± 2%, *p* < 0.001), PX-12 was more effective than ERW1041 (Figure 1A). At these concentrations, PX-12 significantly reduced enzymatic modification of gliadin by more than 80%.

### 2.2. PX-12 Inhibits TG2 in Caco-2 Cells More Effectively Than ERW1041

In order to investigate the inhibition of cellular TG2 by both inhibitors, we used whole cell lysates of differentiated Caco-2 cells (10–14 days after confluency). First, we addressed the question whether reduction of TG2 in the cell lysate can increase cross-linking of the TG2-substrate 5BP. For this purpose, we used dithiothreitol (DTT) as a well-known and strong reducing agent. Treatment of protein lysates with DTT for 1 h significantly increased TG2 activity at a concentration of 10 mM (395 ± 21%, *p* < 0.001, Figure 1B). Additionally, we demonstrated the effect of stimulating Caco-2 cells with IFN-γ, the key cytokine of the Th1-driven inflammation in CD. After treatment with 500 IU/mL IFN-γ for 48 h, TG2 activity was significantly increased (760 ± 259%, *p* < 0.05, Figure 1B). Next, protein lysates were incubated with the substrate 5BP and ERW1041 or PX-12 overnight. Both substances displayed significant, dose-dependent inhibition of cellular TG2 activity (Figure 1C). Again, PX-12 was more effective than ERW1041 (12 ± 7% vs. 45 ± 19%, *p* < 0.05, Figure 1C).

### 2.3. PX-12 Inhibits TG2 in the Duodenal Lamina Propria of CD Patients

Next, we addressed the question whether PX-12 inhibits TG2 in the lamina propria by using an ex vivo approach with biopsies from patients with active CD. For this purpose, we produced tissue sections of native duodenal biopsies from CD patients. To visualize TG2 activity and protein levels, samples were incubated with 5BP as TG2 substrate and a primary antibody directed against TG2. We did not observe any TG2 activity in the cryopreserved tissue sections, although the protein expression was adequate (Figure 2A). Hence, we added the reducing agent DTT to the incubation buffer. Increasing amounts of DTT enhanced TG2 activity in the cryopreserved tissue in a dose-dependent manner (Figure 2A,B).

To investigate the inhibitory effect of PX-12 and ERW1041, we treated the tissue sections with different concentrations of the inhibitors prior to the incubation with 5BP. Treatment with ERW1041 reduced TG2 activity at both tested concentrations of 1 µM (0.94 ± 0.26 vs. 0.58 ± 0.27, *p* < 0.05) and 100 µM (0.94 ± 0.26 vs. 0.31 ± 0.17, *p* < 0.001, Figure 2C,D). Treatment with PX-12 also reduced TG2 activity, which was significant at a concentration of 100 µM (0.91 ± 0.28 vs. 0.44 ± 0.21, *p* < 0.05, Figure 2E,F). We did not observe a significant difference between the two inhibitors.

### 2.4. PX-12 and ERW1041 Do Not Affect Cell Viability up to a Concentration of 100 µM

Prior to testing the effect of the inhibitors on Caco-2 cell monolayers, we investigated the cytotoxicity by using a resazurin-based cell viability assay. For this purpose, confluent Caco-2 cells were incubated with the indicated concentrations of the inhibitors for 24 h in medium. Both substances did not affect cell viability up to a concentration of 100 µM. At concentrations of 200 µM and 500 µM, however, ERW1041 and PX-12 significantly reduced cell viability (Figure 3A). This showed that both drug candidates were tolerated by intestinal epithelial cells in a broad range up to 100 µM.

### 2.5. Inhibition of Extracellular TG2 on Caco-2 Cell Monolayers

Next, we investigated whether ERW1041 and PX-12 were able to inhibit TG2-mediated cross-linking of 5BP to the cell surface of Caco-2 cells. TG2 activity on the cell surface of unpermeabilized Caco-2 cells appeared to be low. Hence, we tested the effect of a 48 h stimulation with 1000 IU/mL IFN-γ, a recognized inducer of TG2 expression. On the protein level, treatment with IFN-γ resulted in an almost sevenfold increase in TG2 expression (*p* < 0.0001, Figure 3B,C). In line with this, extracellular TG2 activity in non-permeabilized Caco-2 cells increased by about 4 times after IFN-γ stimulation (control 4592 ± 1419 vs. 17556 ± 5103 fluorescence counts, *p* < 0.001, Figure 3D). To further address the effect of the oxidative regulation on TG2, we treated naïve and IFN-γ-stimulated Caco-2 cells with 1 mM DTT for 3 h in the presence of 5BP. In non-stimulated Caco-2 cells, this treatment led to a significant increase in TG2-mediated crosslinking (141 ± 17%, *p* < 0.01, Figure 3E). However, there was no statistically significant effect of DTT in IFN-γ-stimulated Caco-2 cells (123 ± 24%, Figure 3E). 

To reveal the effect of different doses of ERW1041 on cell surface TG2 activity, we incubated Caco-2 cells with 5BP for 3 h in the presence of ERW1041. The competitive inhibitor reduced TG2 activity at concentrations of 10 µM to 100 µM, leading to a reduction of TG2 activity by about 32% (*p* < 0.01) and 70% (*p* < 0.0001), respectively (Figure 3F). When applying PX-12 at concentrations ranging between 10 nM and 100 µM, however, we did not observe any significant inhibition of TG2 (Figure 3G).

### 2.6. Inhibition of TG2 Reduces the Transepithelial Passage of P56-88 but Not of P31-43

The role of TG2 in epithelial transport was studied by following the translocation of two specific fluorochrome-conjugated gliadin peptides across an intestinal cell monolayer. Caco-2 cells were grown on semi-permeable transwell supports until reaching a defined density. Then, the promofluor-labeled immunogenic peptide P56-88, containing the repetitive target sequence for TG2-mediated deamidation (PQLP) and the peptide P31-43, known to exert direct inflammatory effects, were applied to the apical compartment. Their translocation from the upper to the lower compartment was determined by fluorometry (Figure 4A). Both peptides were detected in the basal compartment after 3 h of incubation. To examine the mechanisms for uptake and transport, we examined the effect of MBCD (methyl-β-cyclodextrin), a recognized inhibitor of endocytosis. Incubation of P31-43 in the presence of MBCD significantly reduced the transepithelial passage of this peptide (58 ± 7%, *p* < 0.01, Figure 4B). In contrast, transepithelial transport of P56-88 was not affected by MBCD (89 ± 13%, n.s., Figure 4B). In addition, transport of P31-43 remained unaffected by the TG2-specific competitive inhibitor ERW1041 at both tested concentrations (1 µM and 100 µM). However, ERW1041 significantly inhibited the transepithelial passage of immunogenic P56-88 at 1 µM (33 ± 22%; *p* < 0.05) and 100 µM (24 ± 16%; *p* < 0.01) (Figure 4B). PX-12, on the other hand, had no significant effect on the epithelial translocation of P56-88 (Figure 4B).

To obtain more detailed insight into the epithelial transport of immunogenic P56-88, we applied confocal microscopy. Confluent Caco-2 cells were incubated with the peptide for 1 h in the presence of ERW1041 or PX-12. In line with our fluorometric data, ERW1041 reduced the uptake of P56-88 in a dose-dependent manner (Figure 4C,D). Strikingly, peptide uptake was nearly absent in the presence of the higher concentration of ERW1041 (100 µM). Treatment with PX-12, however, did not affect the uptake of P56-88 by Caco-2 cells (Figure 4C,E). To further confirm the localization of P56-88, we performed z-stack imaging and 3D reconstruction of the confocal images, showing that P56-88 localized in the supranuclear cell region (Figure 4F). 

## 3. Discussion

Inhibition of TG2 is currently regarded as one of the most promising strategies for a non-dietary treatment of CD. In the past decades, the effects of competitive amine inhibitors (5BP, monodansyl cadaverine and cystamine) and irreversible inhibitors targeting the active-site cysteine (R281, R283, KCC009, ERW1041 and ZED1227) have been investigated [23,27,28]. Due to the lack of an adequate animal model, most studies used in vitro and ex vivo assays [19,29,30]. 

In addition to the inhibition by targeting the active site of TG2, an allosteric regulatory mechanism has been demonstrated lately. TG2 is activated by the reduction of a disulfide bond (cys 370 and cys 371), whereas oxidation inhibits the enzyme [24,25,31]. In our previous study, we identified the exogenous oxidative molecule PX-12 as a potent inhibitor of TG2 in vitro [26]. We thus aimed to further investigate the inhibitory potential of this molecule, as well as the established irreversible inhibitor ERW1041. 

We demonstrated that PX-12 was more effective than ERW1041 in inhibiting purified TG2 and TG2 from Caco-2 cells. Both inhibitors also reduced TG2 activity within the lamina propria of duodenal biopsies from CD patients. To further examine the applicability of both substances, we analyzed their effect on the viability of Caco-2 cells. Here, we did not observe any negative effect at concentrations up to 100 µM. Other studies reported on cytotoxic effects of PX-12 at lower concentrations (10 µM). In those studies, however, viability was determined in tumor cell lines and not in differentiated intestinal epithelial cells [32,33]. 

When investigating the effect of both inhibitors on the extracellular TG2 activity on an intact Caco-2 cell monolayer, ERW1041 reduced TG2-mediated cross-linking in a dose-dependent manner, whereas PX-12 failed to inhibit TG2. This might be explained by rapid degradation or reduction of PX-12 by living cells [34,35]. This is in line with observations in clinical trials, where PX-12 plasma levels were very low when given intravenously due to rapid binding to plasma components [34,36]. 

Since TG2 is expressed on the surface of epithelial cells and plays a role in transcytosis, we aimed to investigate the effect of TG2 inhibitors on the epithelial uptake of gliadin peptides. First, we confirmed the results from Caputo et al. and Zimmermann et al. by showing that the translocation of P31-43, but not of P56-88, was inhibited by the endocytosis inhibitor MBCD [21,37]. We demonstrated that the cellular uptake and transcellular passage of immunogenic P56-88 depend on TG2, since ERW1041 significantly inhibited both processes. Interestingly, ERW1041 did not inhibit transport of P31-43. This partially contradicts previous research on the uptake of P56-68 [38]. Rauhavirta et al. and Lebreton et al. demonstrated that IgA deriving from CD sera increased transepithelial passage of immunogenic (P57-68) and toxic gliadin peptides (P31-43, P31-49), which could be prohibited in the presence of the site-specific TG2 inhibitor R281 [20,39]. Caputo et al. did not show an effect of the competitive inhibitors cystamine and monodansyl cadaverine on gliadin peptide uptake (P31-43 and P57-68), but antibody-mediated inhibition of TG2 (Clone CUB7402) reduced cellular uptake of both peptides [21]. In our study, we did not observe an effect of TG2 inhibition on the transport of P31-43 in the absence of sIgA. Still, we found that irreversible inhibition of TG2 by ERW1041 significantly diminishes the transepithelial transport of the potentially immunogenic peptide P56-88. This observation might be explained by a higher potency and specificity of ERW1041 compared to inhibitors used in previous studies. Additionally, the application of the higher-molecular-weight peptide P56-88, which contains the repetitive target sequence for TG2 (molecular weight P56-88~4 kDa, P57-68~1.5 kDa), might also explain the differing results. Nonetheless, our data indicate that inhibition of TG2 may prevent the transepithelial passage of immunogenic gliadin peptides, which further emphasizes the potential of this therapeutic approach in CD.

Even though our results underline the significance of TG2 inhibition, further research is needed to investigate the effects of those inhibitors on gliadin-induced inflammation. In recent studies, these inflammatory mechanisms were only partially mitigated by inhibition of TG2. In particular, Molberg et al. found that cystamine prevented T-cell stimulation upon gliadin incubation in only 50% of CD patients [30]. Additionally, Maiuri et al. reported that inhibition of TG2 by R283 did not prevent actin rearrangement after treatment with P31-43 [19]. Furthermore, this inhibitor did not mitigate epithelial infiltration of CD8^+^ T cells and mucosal damage [19], which could be explained by direct toxic and inflammatory effects of gliadin peptide P31-43, irrespective of TG2 activity. However, novel potent and specific inhibitors of TG2, such as ERW1041, might also affect these mechanisms.

In addition, TG2 activity is associated with the development of CD-associated autoimmune diseases, such as type I diabetes, autoimmune thyroiditis and multiple others [40]. Inhibition of TG2 might reduce the production of so-called “neo-epitopes” by trans- and deamidation of exogenous and endogenous proteins. Henceforth, TG2 inhibitors might also prevent the development of autoantibodies and secondary autoimmunity [41,42].

In summary, our data demonstrate that the oxidative inhibition of TG2 is more effective than competitive inhibition on an in vitro and ex vivo level. However, in intestinal cells, PX-12 did not inhibit TG2 activity or gliadin peptide transport, possibly due to rapid inactivation or degradation. The competitive inhibitor ERW1041 reduced TG2 activity in all tested applications and also inhibited epithelial uptake of the immunogenic gliadin peptide P56-88. This indicates that inhibition of TG2 might not only reduce deamidation of immunogenic epitopes, but may also prevent the transepithelial passage of potentially immunogenic gliadin peptides, irrespective of the presence of sIgA.

Henceforth, our results further strengthen the applicability of TG2 inhibitors as a non-dietary treatment option in CD.

## 4. Materials and Methods

### 4.1. Patients’ Characteristics

All duodenal biopsies were obtained during clinically indicated upper gastrointestinal endoscopy. The study was approved by the local ethics committee (reference number 119/16), and written informed consent was obtained from every patient. At the time of endoscopy, all patients were on a gluten-containing diet. Diagnosis of CD was made according to current guidelines of the European Society for Paediatric Gastroenterology, Hepatology and Nutrition (ESPGHAN) [43,44]. Patients’ characteristics are listed in Table 1.

### 4.2. Cell Culture

Human intestinal epithelial cells (Caco-2) were cultured at 37 °C with 5% CO_2_ and 95% humidity in DMEM supplemented with 1% penicillin-streptomycin, 1% essential amino acids, 1% sodium pyruvate and 10% heat-inactivated fetal bovine serum. The culture medium was changed every two to three days, and cells were passaged at 80% confluence using trypsin-EDTA. Cells were seeded at a density of 4 × 10^4^ per cm^2^ onto tissue-culture-treated plates. Cells were used at passages 20–50 and 7–14 days after confluence.

### 4.3. In Vitro Transamidation of Digested Pepsin-/Trypsin-Digested Gliadin

A tissue-culture-treated 96-well plate (83.3924, Sarstedt, Nümbrecht, Germany) was coated for 1 h at 37°C with 10 nM TG2 (T034, Zedira, Darmstadt, Germany) in 50 mM Tris, 1 mM EDTA, 5 mM CaCl2 (pH 7.5) buffer containing 10 mM DTT. After three washes with PBS, the plate was blocked with 2% bovine serum albumin (BSA, Carl Roth GmbH, Karlsruhe, Germany) in PBS for 30 min at room temperature. Then, incubation with 10 µg/mL of pepsin-/trypsin-digested gliadin (PTG, T004, Zedira) ± inhibitors was performed for 1 h at 37 °C. After three washes with PBS, the primary antibody directed against gliadin (1:1000, clone XGY4, Zedira) was applied in BSA 2% at 4 °C overnight on an orbital shaker. The next day, the secondary antibody directed against mouse IgG (1:1000; sc-516102, Santa Cruz Biotechnology, Dallas, TX, USA) was incubated in BSA 2% for 1 h at room temperature on an orbital shaker. After three washes with phosphate buffered saline (PBS), the HRP-substrate TMB (tetramethylbenzidine, Sigma-Aldrich, Darmstadt, Germany) was added (100 µL per well). Photometric quantitation at 655 nm was performed after 15 min using a Clariostar Plus (BMG Labtech, Ortenberg, Germany) microplate reader.

### 4.4. Transglutaminase Activity in Cell Lysates of Caco-2 Cells

The method to investigate TG2 activity from whole cell lysates was adapted from Lin et al. [45]. In brief, 20 µg protein per cavity on the 96-well plate were incubated in 50 mM Tris, 1 mM EDTA, 5 mM calcium chloride (pH 7.5) buffer in the presence of 500 µM 5BP (5-biotinamidopentylamine, Thermo Fisher Scientific, Langenselbold, Germany) for the indicated time. After incubation, the plate was washed three times with PBS + 0.05% Tween. After blocking with 5% BSA in PBS for 1 h, the plate was incubated with 2 µg/mL streptavidin-Alexa Fluor 488 (Thermo Fisher Scientific) in 5% BSA. After 3 washing steps with PBS + 0.05% Tween, fluorometric evaluation was performed using a Clariostar Plus (BMG Labtech) microplate reader (no well scan mode, excitation 488 nm, emission 535 nm, fixed gain: 2000).

### 4.5. TG2 Activity in Duodenal Biopsies

For immunofluorescence microscopic investigation of transglutaminase activity, only biopsies with minor macroscopic inflammatory signs were used. After endoscopy, biopsies were immediately washed three times in PBS and cryopreserved in polyvinylpyrrolidone sucrose at 4 °C overnight. Biopsies were mounted on specimen holders using Tissue-Tek O.C.T (Sakura Finetek, Staufen im Breisgau, Germany) compound and frozen in liquid nitrogen. For evaluation of transglutaminase activity by the post-embedding procedure, 400 nm cryosections were obtained using a cryo-ultramicrotome (Leica EM UC6, Leica, Wetzlar, Germany) and mounted on class cover slides. After three brief washes with PBS, tissue sections were incubated with the indicated concentrations of the inhibitors for 1 h in tris buffer. Then, the TG2 substrate 5BP was incubated for 1 h at 37 °C in tris buffer containing DTT at the indicated concentration. Slides were washed three times in PBS, fixed in 4% paraformaldehyde (10 min), permeabilized with 0.5% Triton (10 min) and incubated with blocking solution (0.1% cold water fish skin gelatin + 5% BSA + 5% goat serum, 10 min). Afterwards, a rabbit polyclonal antibody directed against human TG2 (Kan5, kindly provided by Dr T. Mothes, Institute of Laboratory Medicine, University Hospital Leipzig, Germany) was applied at 1:200 dilution in blocking solution overnight at 4 °C. After three washing steps (each 5 min) with PBS, Alexa Fluor-555-conjugated secondary antibody (goat-anti-rabbit, 1:200, Thermo Fisher Scientific) and Alexa Fluor-488-conjugated streptavidin (2 µg/mL, Thermo Fisher Scientific) were incubated at room temperature in the dark for 1 h. After three washes with PBS, samples were covered in ProLong Gold antifade mountant (Thermo Fisher Scientific) and sealed with clear nail polish. Sealed slides were stored in the dark at 4 °C until microscopy.

Imaging was done using a confocal laser scanning microscope (TE2000-E, Nikon, Langen, Germany) and a 60× Plan Apo (NA 1.41) immersion oil objective. The detailed acquisition settings and the corresponding macro (Image J [46]) for image processing can be accessed in the Appendix A methods (S1). Five images per patient where quantified. For quantitation, mean fluorescence intensity of TG2 activity (488 nm channel) was normalized to TG2 expression (555 nm channel) within the intestinal lamina propria.

### 4.6. Cell Viability of Caco-2 Cells

The cell viability of Caco-2 cells was examined using PrestoBlue HS (A13261, (Thermo Fisher Scientific). In brief, Caco-2 cells (7–8 days after confluency) were incubated with PX-12 or ERW1041 at the indicated concentrations for 24 h in full medium. As a positive control, cells were incubated with 4% PFA for 10 min. After washing with PBS, cells were incubated with PrestoBlue HS diluted 1:10 in culture medium for 2 h at 37 °C. Fluorometric examination was conducted using a Clariostar Plus (no well scan mode, bottom optic measurement, excitation 545 nm, emission 600 nm, fixed gain: 800). Every experiment was performed in duplicates with at least three technical replicates. Normalization was performed against conditions, where cells were not treated with TG2 inhibitors.

### 4.7. Western Blotting

Caco-2 cells were incubated with 500 IU/mL IFN-γ for 48 h and harvested on day 12 after confluency. First, cells were washed two times with cold PBS, scraped off the plate and centrifuged (60 s at 14,100× *g*). The cell pellet was resuspended in RIPA lysis buffer supplemented with phenylmethylsulfonyl fluoride, protease inhibitor and sodium orthovanadate (1% each), incubated for 20 min on ice and centrifuged (20 min at 14,100 gat 4 °C) afterwards. Supernatant was collected and cell debris was discarded. Protein quantitation was performed using the Pierce BCA protein assay kit (Thermo Fisher Scientific). After denaturation in Laemmli buffer containing 2-mercaptoethanol for 5 min at 95 °C, 10 µg of the protein extract per lane were loaded on a 12% SDS-gel. Electrophoresis was done for 1 h at 40 mA. Protein transfer onto PVDF membrane (0.2 µm pore size) was done using a Transblot Turbo system (Bio-Rad Laboratories) for 30 min (25 V, 1 A). The membrane was blocked for 1 h with 5% non-fat dry milk in PBS + 2.5% Tween. The primary antibodies (anti-TG2 (1:500): CUB7402, Abcam, Cambridge, UK; anti-GAPDH (1:5000): MAB374, Merck Millipore, Darmstadt, Germany) were incubated at 4 °C in blocking buffer overnight. After three washes with PBS + 2.5% Tween, the membrane was incubated with horseradish peroxidase-conjugated secondary antibody (1:1000, sc-516102, Santa Cruz Biotechnology, Dallas, TX, USA) for 1 h. The membrane was washed again three times with PBS + 2.5% Tween and developed using the SuperSignal West Femto kit (Thermo Fisher Scientific) for 5 min. Image processing and quantitation were performed using a ChemiDoc XRS+ imager and Image Lab software (BioRad Laboratories, Feldkirchen, Germany).

### 4.8. Inhibition of Extracellular TG2 Activity in Caco-2 Cells

The method was adapted from Yi et al. [24]. To induce TG2 expression, Caco-2 cells were stimulated with IFN-γ 1000 IU/mL for 48 h. Then, cells were washed three times with warm PBS and incubated with the substrate 5BP for 3 h at 37 °C in 50 mM Tris and 1 mM EDTA (pH 7.5). The substances ERW1041, PX-12 and DTT were added to this solution when indicated. After incubation, cells were washed three times with PBS, fixed with 4% paraformaldehyde for 10 min and blocked with 5% BSA in PBS at 4 °C overnight. Then, cells were incubated at room temperature with streptavidin-Alexa488 (Thermo Scientific, 4 µg/mL) in 5% BSA in the dark for 1 h, followed by three washes with PBS. Fluorometric quantitation was conducted using a FLUOstar Optima microplate reader (BMG Labtech, filter settings: excitation 485 nm, emission 525 nm). Manual gain adjustment was performed, setting the well with the most intense signal as 95%. In all experiments, data were background subtracted against a control without 5BP substrate.

### 4.9. Epithelial Transport of Gliadin Peptides in Caco-2 Cells

Transepithelial transport of immunogenic and toxic gliadin peptides were examined in Caco-2 cells 7–14 days after confluency. Caco-2 cells were seeded on semi-permeable transwell supports (ThinCert, culture surface: 33.6 mm², pore size: 0.4 µm) in 24-well plates (Greiner Bio One, Pleidelsheim, Germany). Prior to the transport experiments, the integrity of the cell monolayer was tested by measuring the transepithelial electric resistance (TEER) using a volt-ohm meter (Millipore, Schwalbach, Germany). Monolayers with insufficient TEER (<350 Ω/cm^2^) were omitted. The synthesis and purification of promofluor-conjugated gliadin peptides were described elsewhere [37]. Cells were either incubated with the immunogenic peptide P56-88 (LQLQPFPQPQLPYPQPQLPYPQPQLPYPQPQPF) or the toxic gliadin peptide P31-43 (LGQQQPFPPQQPY), 7 µg/mL each, in Hank’s balanced salt solution (HBSS) for 3 h. To investigate the role of TG2 on the transport of gliadin peptides, incubation was performed in the presence of ERW1041 and PX-12. After 3 h, basal and apical supernatants were collected, and the relative fluorescence of the basal compartment was quantified using a FLUOstar Optima (BMG Labtech) microplate reader. Background subtraction was performed against conditions, where no labeled peptides were used. Normalization was performed against conditions, where no inhibitors were used.

For microscopic evaluation, Caco-2 cells were seeded on polymer-covered µ-slides (80826, Ibidi) and incubated for 1 h with labeled gliadin peptides in the presence of TG2 inhibitors. After three washes with PBS, cells were fixed with 4% paraformaldehyde. Nuclear staining was performed for 10 min using Hoechst fluorescent stain (1:1000 in PBS). Imaging was done using a confocal laser scanning microscope (TE2000-E, Nikon) and a 60× Plan Apo (NA 1.41) immersion oil objective. Processing of z-stacks, including 3D reconstruction, was performed using ImageJ [46]. For quantitation, z-stack images were taken with the same microscope settings. Using ImageJ, maximum intensity projections were obtained of the z-stack images, and the fluorescence intensity was measured within the whole image.

### 4.10. Statistics

Statistical analysis was performed using GraphPad Prism 9 (GraphPad Prism Software Inc., San Diego, CA, USA). Results were tested for normal distribution. Student’s unpaired two-tailed *t*-test (with Welch correction where appropriate) or one-way ANOVA with Dunnett’s multiple comparison test were used where appropriate. *p* values less than 0.05 were considered significant.

## Figures and Tables

**Figure 1 ijms-24-04795-f001:**
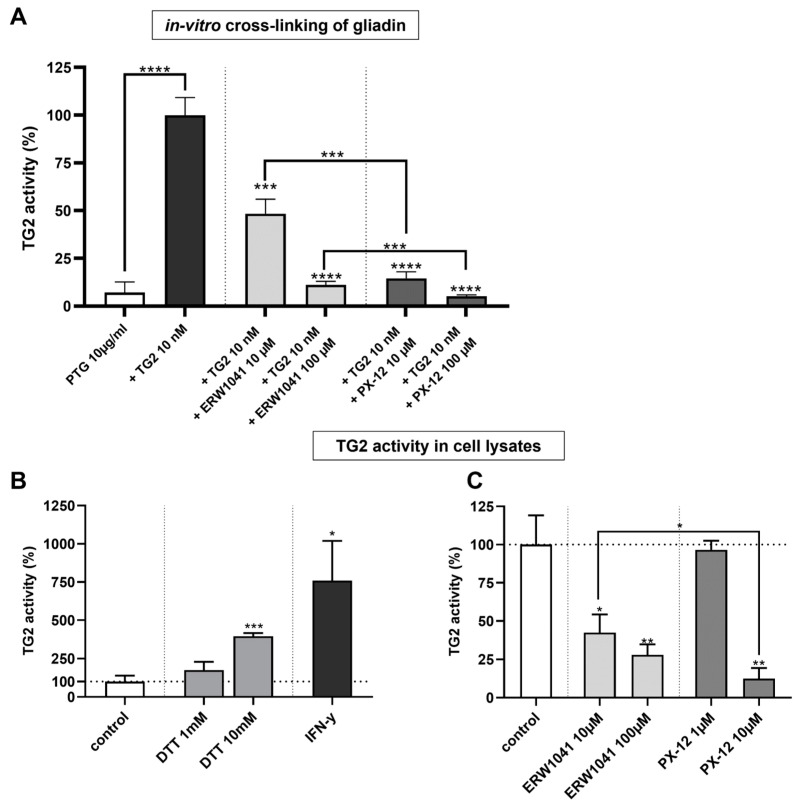
PX-12 inhibits recombinant and cellular TG2 more effectively than ERW1041. (**A**) TG2-mediated cross-linking of PTG was quantified by fluorometry. Normalization was performed against a condition, where the plate was coated with TG2 and no inhibitors were used. Background subtraction was performed against conditions where neither PTG nor TG2 were applied. *n* = 4 with three technical replicates per experiment. (**B**) TG2 activity in Caco-2 cell lysates was determined by cross-linking of 5BP (1 h). Reduction of protein lysate by DTT or treatment of Caco-2 cells with IFN-γ for 48 h significantly increased TG2 activity. Background subtraction was performed against conditions where no protein lysate was applied. *n* = 3 with three technical replicates per experiment. (**C**) Cell lysates were incubated overnight in the presence of inhibitors. Inhibition of cellular TG2 by PX-12 is more effective than competitive inhibition by ERW1041. Background subtraction was performed against conditions where no protein lysate was applied. *n* = 3 with three technical replicates per experiment. All data are shown as mean ± SD. Statistical significance was tested with Student’s *t*-test (with Welch correction where appropriate). * *p* < 0.05, ** *p* < 0.01, *** *p* < 0.001, **** *p* < 0.0001.

**Figure 2 ijms-24-04795-f002:**
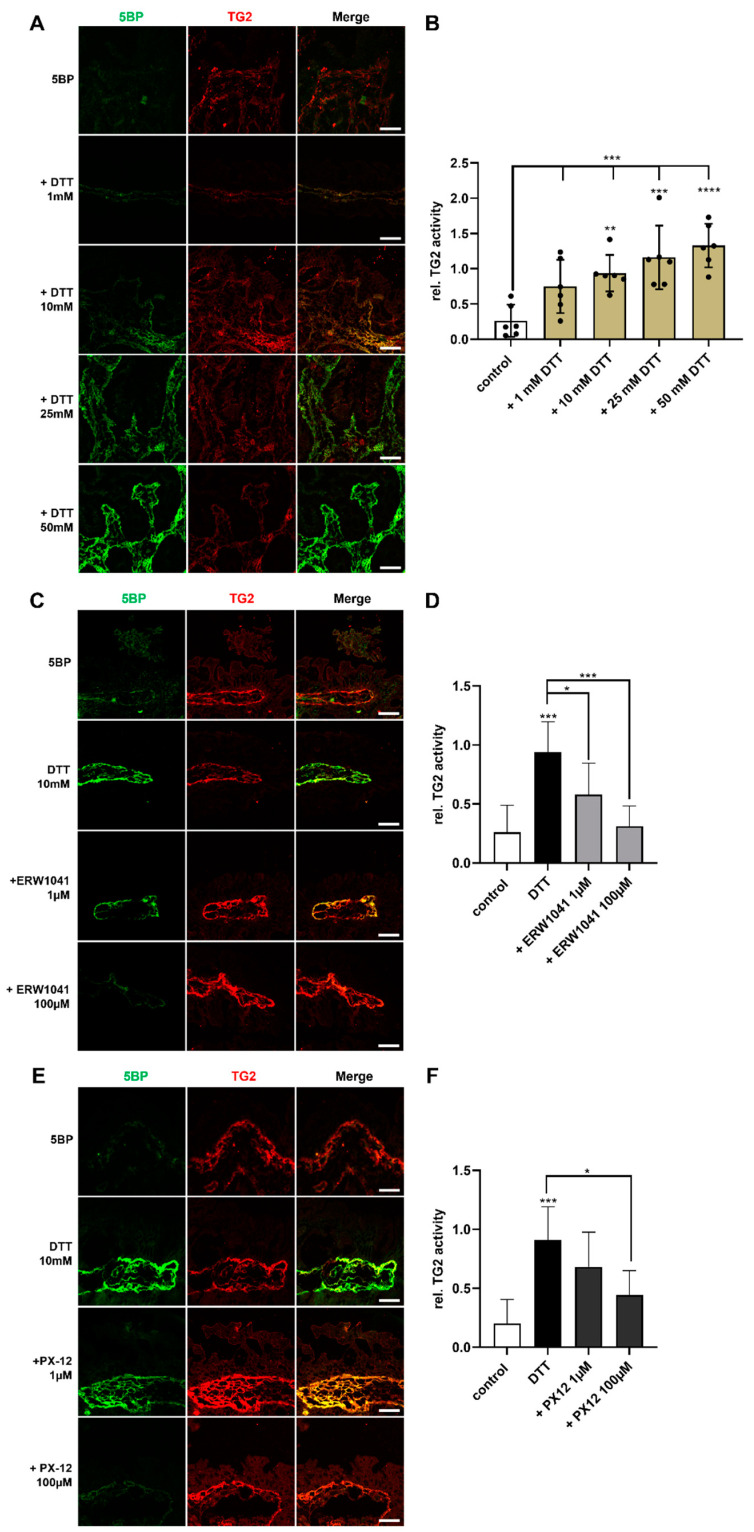
PX-12 inhibits TG2 in the intestinal lamina propria. (**A**) Unfixed cryosections (400 nm) of duodenal biopsies were incubated with 5BP in the presence of increasing concentrations of DTT. Increasing amounts of DTT activate TG2 in the lamina propria. (**B**) TG2-activity (green) was normalized against TG2-expression (red) and quantified using ImageJ. Higher amounts of DTT led to increased TG2 activity. *n* = 6; 5 images per patient were quantified. (**C**–**F**) Cryosections were treated with ERW1041 and PX-12 prior to the incubation with 5BP. Both inhibitors significantly reduced TG2 activity in the lamina propria. *n* = 6; 5 images per patient were quantified. All data are shown as mean ± SD. Statistical significance was tested with one-way ANOVA and Dunnett’s multiple comparison test. * *p* < 0.05; ** *p* < 0.01; *** *p* < 0.001; **** *p* < 0.0001. Scale bars: 20 µm.

**Figure 3 ijms-24-04795-f003:**
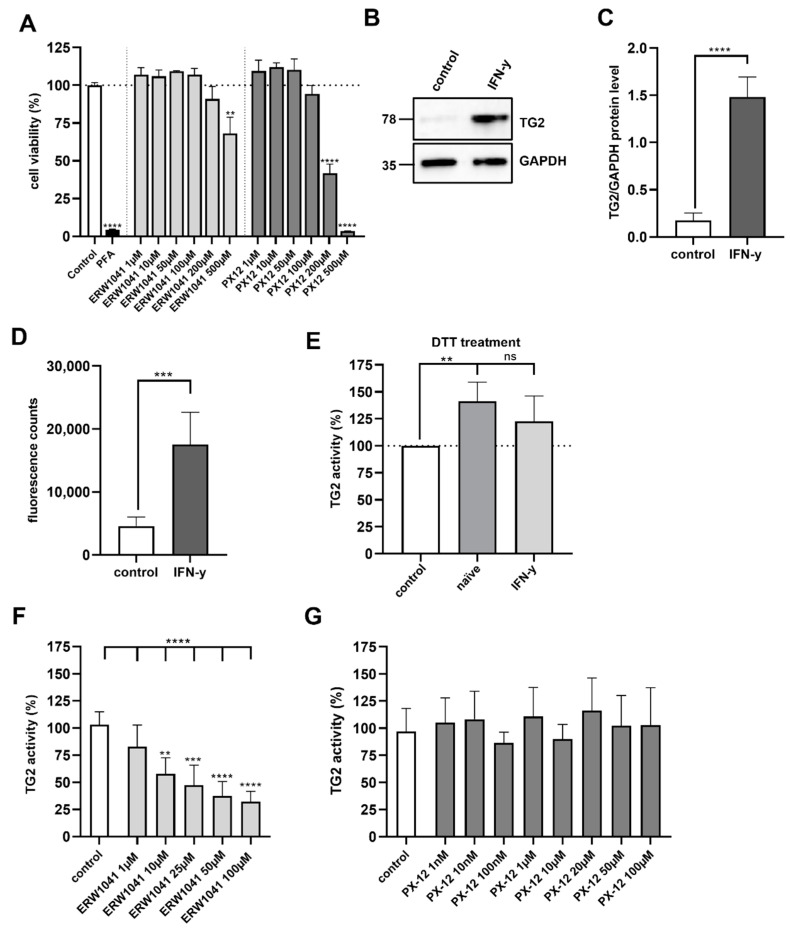
Evaluation of TG2 inhibition on confluent Caco-2 cell monolayers. (**A**) Cell viability of Caco-2 cells was tested after 24 h treatment with ERW1041 and PX12 at the indicated concentrations. *n* = 3 with three technical replicates per experiment. (**B**) Protein expression of TG2 in Caco-2 cells was analyzed by Western blotting after treatment IFN-γ. GAPDH served as loading control. (**C**) Relative quantities were normalized to GAPDH levels in whole cell lysates. IFN-γ stimulation significantly increased TG2 levels in differentiated Caco-2 cells (*n* = 4). (**D**) TG2 activity in confluent Caco-2 cell monolayers was investigated by fluorometry after stimulation with IFN-γ (1000 IU/mL) for 48 h. IFN-γ treatment significantly increased TG2 activity on the cell surface. *n* = 6 with three technical replicates per experiment. (**E**) Naïve and IFN-γ-stimulated Caco-2 cells were treated with 1 mM DTT in the presence of 5BP. Crosslinking of 5BP was evaluated by fluorometry. In non-stimulated Caco-2 cells, DTT treatment significantly increased TG2 activity, but not in IFN-γ-stimulated Caco-2 cells. *n* = 4 with three technical replicates per experiment. (**F**) Caco-2 cells were incubated with 5BP in the presence of different amounts of ERW1041 for 3 h. TG2-mediated crosslinking was quantified by fluorometry. ERW1041 inhibited extracellular TG2 in a dose-dependent manner. *n* = 4 with three technical replicates per experiment. (**G**) PX-12 did not inhibit TG2 activity on confluent monolayers of Caco-2 cells at the tested concentrations. *n* = 4 with three technical replicates per experiment. All data are shown as mean ± SD. Statistical significance was tested with Student’s *t*-test (with Welch correction where appropriate) and one-way ANOVA with Dunnett’s multiple comparison test (**E**), ** *p* < 0.01, *** *p* < 0.001, **** *p* < 0.0001; ns = not significant.

**Figure 4 ijms-24-04795-f004:**
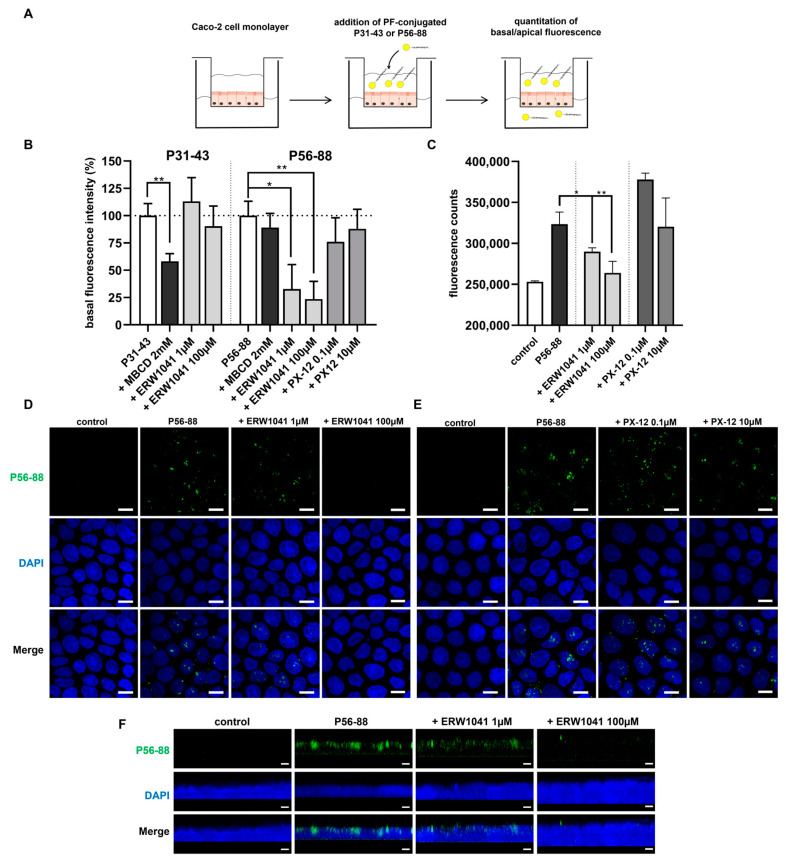
Inhibition of epithelial transport of P56-88 by ERW1041. (**A**) Confluent Caco-2 cell monolayers were incubated with promofluor-conjugated gliadin peptides P31-43 or P56-88 in the presence of ERW1041 or PX-12. Epithelial translocation of gliadin peptides was quantified by fluorometry of apical and basal supernatant. (**B**) Relative basal fluorescence intensity was quantified after a 3 h incubation of conjugated gliadin peptides in the presence of inhibitors. ERW1041 reduced epithelial translocation of P56-88. *n* = 3 with three technical replicates per experiment. (**C**) Confocal microscopical quantitation of epithelial uptake of P56-88 after 1 h in the presence of ERW1041 and PX-12. *n* = 3 with three images per experimental condition. (**D**,**E**) Confocal images of P56-88 uptake by Caco-2 cells in the presence of different amounts of ERW1041 (D) and PX-12 (**E**). ERW1041 inhibited epithelial uptake of P56-88 in a dose-dependent manner. (**F**) 3D reconstruction of z-stack images. P56-88 was localized in the supranuclear cell region after 1 h of incubation. ERW1041 inhibited the uptake of P56-88. All data are shown as mean ± SD. Statistical significance was tested with Student’s *t*-test. * *p* < 0.05, ** *p* < 0.01.

**Table 1 ijms-24-04795-t001:** Patients’ characteristics.

Patient	Age	Sex	Anti-TG2-IgA (IU/mL)	Marsh
1	14	m	200	3a
2	7	m	200	3a
3	5	f	134	3a
4	3	m	200	3a
5	16	f	200	3a-b
6	10	m	200	3a-b

## Data Availability

The data presented in this study are available on request from the corresponding author.

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
