# Peer review of "Inhibition of Transglutaminase 2 as a Therapeutic Strategy in Celiac Disease—In Vitro Studies in Intestinal Cells and Duodenal Biopsies"

_ijms, 2023, doi:10.3390/ijms24054795_

Round 1

Reviewer 1 Report

In this study, the authors investigated the effect of the small oxidative molecule PX-12 (a potential drug candidate in CD) and the established active-site directed inhibitor ERW1041 on TG2 activity and epithelial transport of gliadin peptides.

They analyzed TG2 activity using immobilized TG2, Caco-2 cell lysates, confluent Caco-2 cell monolayers, and duodenal biopsies from CD patients. TG2-mediated cross-linking of pepsin-/trypsin-digested gliadin (PTG) and 5BP (5-biotinamidopentylamine) was quantified by colorimetry, fluorometry and confocal microscopy. Epithelial transport of promofluor-conjugated gliadin peptides P31-43 and P56-88 was analyzed by fluorometry and confocal microscopy. PX-12 reduced TG2-mediated cross-linking of PTG and was significantly more effective than ERW1041. Further, PX-12 inhibited TG2 in cell lysates obtained from Caco-2 cells more than ERW1041. Both substances inhibited TG2 comparably in the intestinal lamina propria of duodenal biopsies. PX-12 did not reduce TG2 activity or gliadin peptide transport in confluent Caco-2 cells. Thus the finding that the TG2-specific inhibitor ERW1041 reduced the epithelial uptake of P56-88 in Caco-2 cells supports the therapeutic potential of TG2 inhibitors in CD.

The study is of interest with potential clinical/therapeutic significance. However, the TG2 inhibition treatment approach is now under investigation in clinical trials to confirm its safety and efficacy in real-life scenarios. Since it has been previously proposed that TG2 mediates secondary autoimmunity in celiac disease patients (Coeliac disease and secondary autoimmunity. Dig Liver Dis. 2002;34:13-5), to further improve the therapeutic impact of TG2-inhibition approach, the authors should further discuss the potential impact of TG2 inhibition in the associated autoimmune phenomena/disorders that are well-known and reported in celiac patients. In fact, since TG2 may cross-link several external and endogenous antigens, several neoantigens and autoantibodies against cytoskeleton (actin), neuronal (ganglioside) and other antigens, are frequently detected, as previously demonstrated (Anti-actin IgA antibodies in severe coeliac disease. Clin Exp Immunol. 2004;137:386-92; Anti-ganglioside antibodies in coeliac disease with neurological disorders. Dig Liver Dis. 2006;38:183-7; Sera of patients with celiac disease and neurologic disorders evoke a mitochondrial-dependent apoptosis in vitro. Gastroenterology. 2007;133:195-206; Anti-tissue transglutaminase antibodies as predictors of silent coeliac disease in patients with hypertransaminasaemia of unknown origin. Dig Liver Dis. 2001;33:420-5.). A paragraph discussing the potential efficacy of TG2 inhibition treatment approach on the secondary autoimmunity observed in celiac patients would further improve the clinical significance of this study.

Author Response

We thank reviewer 1 for taking the time to carefully read and evaluate our manuscript. We appreciate the constructive feedback regarding the impact of TG2-inhibition of secondary autoimmunity in CD. We added a paragraph to the discussion section in which we discuss the potential impact of TG2 inhibition on the development of autoantibodies and secondary autoimmunity.

“In addition, TG2 activity is associated with the development of CD-associated autoimmune diseases such as type I diabetes, autoimmune thyreoiditis and multiple others [40]. Inhibition of TG2 might reduce the production of so-called “neo-epitopes” by trans- and deamidation of exogenous and endogenous proteins. Henceforth, TG2 inhibitors might also prevent the development of autoantibodies and secondary autoimmunity [41,42].”

Reviewer 2 Report

The article is suitable for publication.

Author Response

We thank reviewer 2 for taking the time to critically review our manuscript. Reviewer 2 did not demand any changes of the manuscript.

Reviewer 3 Report

Comments to Authors

The paper titled “Inhibition of transglutaminase 2 as a therapeutic strategy in 2 celiac disease - In vitro studies in intestinal cells and duodenal biopsies” is well written and of interest for researchers who deal with TG2 and its role in the pathogenesis of celiac disease. In this study the authors investigate in vitro end in both Caco2 cells and duodenal biopsies from CD patients the inhibiting capability of PX-12 of tG2 activity, furthermore they tested the effect of PX-12 and of ERW1041 on epithelial transport of gliadin peptides. They showed that oxidative inhibition of TG2 by PX12 is more effective than competitive inhibition by ERW1041 in vitro and in both Caco2 cells and duodenal biopsies. Moreover, PX12 did not inhibit gliadin peptide transport while ERW1041 inhibited transepithelial transport of immunogenic gliadin peptide, P5688, but not of P3143.

Showed data are convincing but I have some doubts about the experiments with duodenal biopsies. Only 3 patients were enrolled with Marsh 3 a-b mucosal lesion, as reported, and they are too few to perform a reliable statistical analysis. A higher number of patients, at least double, would allow for more solid statistical evaluations. The graph B, in figure 2, is a bar graph but I would evaluate a scatter dot plot one to see individual data patient. Furthermore, the immunofluorescence images in figure2A, C, D showed a duodenal mucosa with villa (that even appear to be of normal height) and it is not representative of an atrophic mucosa. Why? The Authors should perform the experiments of inhibition of tissue transglutaminase 2 by Px12 and ERW1041 recurring to Marsh 3b or 3c mucosa and show the images of stained damaged duodenal mucosa. Finally, panels F in figure4 could be larger to better appreciate immunofluorescnt staining. The manuscript is not yet ready for publication in International Journal of Molecular Sciences

Round 2

Reviewer 3 Report

The reviewer thanks the Authors for making the requested changes. The manuscript has been improved and is ready for publication in IJMS